# Nitrogen-rich covalent organic frameworks with multiple carbonyls for high-performance sodium batteries

Ruijuan Shi[1], Luojia Liu[1], Yong Lu [1], Chenchen Wang[1], Yixin Li[1], Lin Li [1], Zhenhua Yan[1] & Jun Chen [1]*

Covalent organic frameworks with designable periodic skeletons and ordered nanopores have attracted increasing attention as promising cathode materials for rechargeable batteries. However, the reported cathodes are plagued by limited capacity and unsatisfying rate performance. Here we report a honeycomb-like nitrogen-rich covalent organic framework with multiple carbonyls. The sodium storage ability of pyrazines and carbonyls and the up-to twelve sodium-ion redox chemistry mechanism for each repetitive unit have been demonstrated by in/ex-situ Fourier transform infrared spectra and density functional theory calculations. The insoluble electrode exhibits a remarkably high specific capacity of 452.0 mAh $g^{-1}$, excellent cycling stability (~96% capacity retention after 1000 cycles) and high rate performance (134.3 mAh $g^{-1}$ at 10.0 A $g^{-1}$). Furthermore, a pouch-type battery is assembled, displaying the gravimetric and volumetric energy density of 101.1 Wh $kg^{-1}_{cell}$ and 78.5 Wh $L^{-1}_{cell}$, respectively, indicating potentially practical applications of conjugated polymers in rechargeable batteries.

[1] Key Laboratory of Advanced Energy Materials Chemistry (Ministry of Education), Renewable Energy Conversion and Storage Center, College of Chemistry, Nankai University, Tianjin 300071, China. *email: chenabc@nankai.edu.cn

Covalent organic frameworks (COFs), which are a class of polymers with designable periodic skeletons and ordered nanopores, have been demonstrated potential applications widely in the fields such as catalysis, semiconductors, proton conduction, and gas capture[1–6]. Moreover, COFs with controllable pore size and redox sites can also be applied in the field of electrochemical energy storage and conversion[7–9]. For efficient battery electrode applications, the skeletons of COFs need contain active groups such as C=O and C=N bonds where O and N atoms could combine with ions (e.g. $Li^+$, $Na^+$, and $K^+$)[10–12]. In addition, the nanopores should be large enough to accommodate ions like $Na^+$ without evident volume expansion and promote facile ions transport[13]. Generally, COFs-based electrode materials show the merits of low cost, environmental friendliness, structural designability, and sustainability[14,15].

In fact, the application of COFs-based cathode materials for rechargeable batteries is just in the beginning[16,17]. For example, Kaskel's group reported a microporous (1.4 nm) bipolar COFs with active triazine rings and inactive benzene rings in the skeletons as an organic electrode for sodium-ion batteries, showing specific capacities of ~200 and 10 mAh g$^{-1}$ at 0.01 and 10 A g$^{-1}$, respectively[18]. Subsequently, Wang's group designed three different types of exfoliated COFs for lithium batteries, exhibiting specific capacities of 145, 210, and 110 mAh g$^{-1}$ for anthraquinone-based, benzoquinone-based, and nitroxyl radical-based COFs, respectively[19]. Recent work revealed that the pentacenetetrone-based π-conjugated COFs with active C=O bonds and inactive linkage groups in the skeletons displayed a discharge capacity of ~120 mAh g$^{-1}$ at a high rate of 5.0 A g$^{-1}$ and a capacity retention of 86% after 1000 cycles at 1.0 A g$^{-1}$ in sodium batteries[20]. The reported COFs-based cathodes for rechargeable batteries are still plagued by low capacity (~200 mAh g$^{-1}$) and inferior rate capability, limiting their further applications[21,22]. The limited capacity of COFs-based cathodes can be ascribed to the introduction of inactive components (e.g., benzene skeleton, boronate esters, and hydrazones), which are used as the linkages to connect active groups within the molecules[23–25]. Additionally, the relatively poor electronic and ionic conductivities of the reported COFs-based electrode materials lead to their inferior rate capability[26,27]. The reported method to improve the rate performance of COFs is combining with conductive carbon materials such as graphene[28–30]. However, the addition of too many carbon materials inevitably decreases the whole energy density of practical batteries. Overall, realizing COFs-based cathode materials with high capacity and high-rate performance is still challenging nowadays.

Herein, we report the design, synthesis and battery application of a COF with triquinoxalinylene and benzoquinone units (TQBQ) in the skeletons via a triple condensation reaction between tetraminophenone (TABQ) and cyclohexanehexaone (CHHO). Benefiting from the absence of inactive linkage groups in the skeletons, the TQBQ-COF electrode shows a high reversible capacity of 452.0 mAh g$^{-1}$ and maintains 352.3 mAh g$^{-1}$ after 100 cycles at 0.02 A g$^{-1}$ in sodium batteries. Furthermore, the introduction of N atoms reduces the energy gap between the lowest unoccupied molecular orbital (LUMO) and the highest occupied molecular orbital (HOMO), resulting in enhanced electronic conductivity (~10$^{-9}$ S cm$^{-1}$) and high ionic conductivity (~10$^{-4}$ S cm$^{-1}$ for the discharged product). As a result, the TQBQ-COF shows a high rate capability of 134.3 mAh g$^{-1}$ at 10.0 A g$^{-1}$. In addition, the insoluble TQBQ-COF electrode exhibits excellent cycling stability with a capacity retention of 96% after 1000 cycles at 1.0 A g$^{-1}$. Moreover, the combination of in/ex-situ Fourier transform infrared (FTIR) spectra and density functional theory (DFT) calculations demonstrate that the pyrazines (C=N) and carbonyls (C=O) are the active sites, and per TQBQ-COF repetitive unit could store twelve $Na^+$ ions, including six $Na^+$ ions within the TQBQ-COF plane and another six $Na^+$ ions outside the plane. The Mulliken charges of the two types of Na atoms and their adjacent N and O atoms can be obtained according to the accurate atomic coordinates of $Na_{12}$TQBQ-COF. Furthermore, the average atomic valences of Na, O, and N in $Na_{12}$TQBQ-COF are calculated to be +1, −1, and −0.5, respectively. In addition, a pouch-type sodium battery with a capacity of 81 mAh is fabricated, showing the way for the application of large batteries.

## Results

**Design and synthesis of TQBQ-COF**. To achieve high capacity and good rate performance, a TQBQ-COF was designed by removing inactive linkage groups in the skeletons and doping heteroatoms. The TQBQ-COF with dual redox sites (C=O and C=N) were synthesized via a solvothermal reaction[31] between TABQ and CHHO at 100 °C for 48 h (Supplementary Fig. 1), followed by heated at 140 °C under Ar bubbling for another 6 h. After further annealing at 200 °C for 5 h under argon atmosphere, the product was achieved as a dark-red powder with a yield of 80%. The facile synthetic process is beneficial for the large-scale preparation of TQBQ-COF. The TQBQ-COF materials consist of multiple carbonyls and pyrazine groups, where carbonyls are used as redox sites and the active pyrazine sites act as the linkage blocks to form the two-dimensional (2D) conjugated framework. As the carbonyls and pyrazine groups are both designed to be redox sites for TQBQ-COF electrode, a theoretical capacity of 515 mAh g$^{-1}$ (based on one repetitive unit marked inside the yellow dotted line in Fig. 1a) could be obtained. DFT calculations are applied to calculate the optimized structures of TQBQ-COF, showing a hexagonal micropore of 11.4 Å and a packing distance of 3.07 Å according to the simulated AB stacking model (Fig. 1b).

**Structural characterizations**. The FTIR spectra of TQBQ-COF exhibits two distinct absorption peaks at 1627 cm$^{-1}$ and 1545 cm$^{-1}$ (Supplementary Fig. 2), which can be assigned to the stretching vibration modes of carbonyls (C=O) and imides (C=N), respectively[28,31]. In addition, the peaks of the heat-treated TQBQ-COF at 1365 cm$^{-1}$ and 3373 cm$^{-1}$ (corresponding to the stretching vibration of C–N and N–H of TABQ-COF) both show obvious decrease in their peak intensity, suggesting the polymerization of the starting reagents. All peaks in the solid-state $^{13}$C NMR spectrum of TQBQ-COF are marked in the indicated groups of the inset chemical structure (Supplementary Fig. 3). The peaks at about 170 ppm and 143 ppm can be assigned to the carbonyl groups and the formation of imide bonds by the Schiff base reaction[32], respectively, further confirming the structure of TQBQ-COF. The proportion of carbon, nitrogen, and oxygen elements account for 53.1, 23.8, and 22.0 wt% from the elemental analysis (Supplementary Table 1). The excess content of oxygen element can be ascribed to the edge groups or small molecules (such as $H_2O$, $CO_2$, and $CH_3OH$) absorbed in the pores.

The crystallinity of TQBQ-COF was investigated by Powder X-Ray Diffraction (PXRD) and high-resolution transmission electron microscopy (HRTEM), as shown in Fig. 1. The strong diffraction peak at 28.24° is attributed to the (002) plane (Fig. 1c), which is related to the interlayer distance of 3.0 ± 0.2 Å between the conjugated TQBQ-COF layers of the simulated AB stacking model (Fig. 1b). The peaks at 13.74° and 19.84° could be assigned to (−220) and (201) facets, respectively. In addition, the peak at 15.69° in PXRD pattern could be assigned to the d-spacing of (0–11) plane, which is in agreement with the pore size of ~5.6 Å of the AB stacking model of TQBQ-COF. Affected by the strong

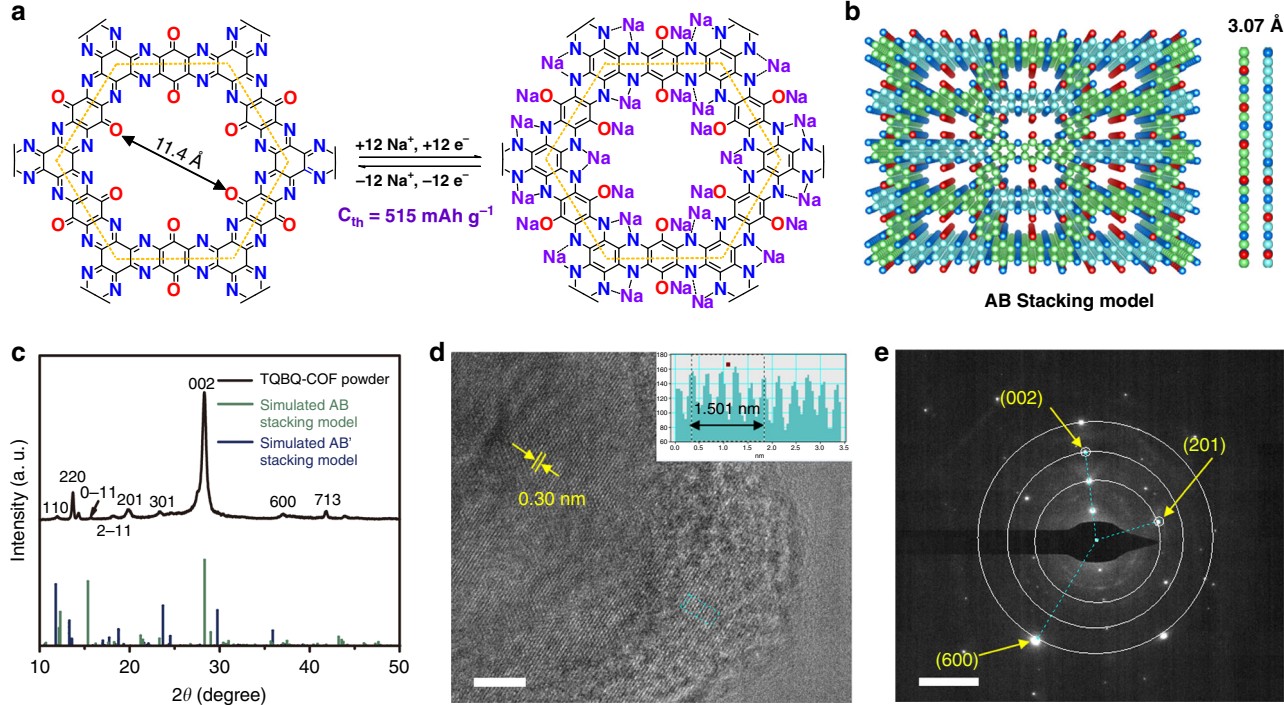

**Fig. 1 Structure and characterizations of TQBQ-COF materials. a** The chemical structure and possible electrochemical redox mechanism of TQBQ-COF with a theoretical capacity of 515 mAh g⁻¹. **b** Top and side views of the schematic AB stacking model of TQBQ-COF layers with a packing distance of 3.07 Å. **c** PXRD pattern and the simulated AB/AB' stacking models of TQBQ-COF powder. **d** High-resolution TEM image of TQBQ-COF with a multilayer stacking space of ~0.30 nm. Scale bar, 5 nm. **e** SAED pattern of (002), (201), and (600) facets of TQBQ-COF from **d**. Scale bar, 5 nm⁻¹.

p–π interaction between the adjacent 2D layers, TQBQ-COF layers tend to contact with each other by an alternative staggered stacking model (AB stacking model, Fig. 1b and Supplementary Fig. 4)[31]. HRTEM image also shows that TQBQ-COF has a moderate crystallinity with periodicities (Fig. 1d), and the obvious lattice fringe distance (0.30 nm) is well referring to the interlayer distance of TQBQ-COF layers. As shown in Fig. 1e, the selected area electron diffraction (SAED) pattern shows the alignment of the hexagonal pore along the (002), (201) and (600) facets of TQBQ-COF with d-spacing of ~3.0 Å, ~4.6 Å, and ~2.4 Å, respectively. Therefore, both the experimental PXRD and HRTEM results match the proposed AB stacking model well. It is noted that a few mismatching peaks are still existing between the experiment PXRD result and the simulation AB stacking pattern in Fig. 1c, which could be ascribed to the existence of the AA/AB' stacking model from a little isotropic powder (Supplementary Fig. 4a). Meanwhile, the HRTEM elemental mappings show uniform distribution of C, N, and O (Supplementary Fig. 5). The TQBQ-COF materials with a few defects and porous structure could combine with Na⁺ well, similar to the sodium storage behavior of disordered soft and hard carbons[33].

X-ray photoelectron spectroscopy (XPS) was performed to further confirm the chemical compositions of TQBQ-COF. The characteristic bands for the K-edge of carbon, nitrogen, and oxygen elements are exhibited without any other impurities from the survey scan XPS spectrum (Supplementary Fig. 6). The C1s spectrum (Supplementary Fig. 7a) mainly displays C=C, C=O, and C=N at 284.5, 287.8, and 287.0 eV, respectively. The binding energy of C=N is close to that of C=O, which can be attributed to the delocalization of lone pair electrons of N and O atoms on the π-conjugated aromatic structure of TQBQ-COF[31]. The peak at 399.6 eV is derived from the C=N of the pyrazine moiety[34] in the N1s spectrum (Supplementary Fig. 7b). All the results confirm the stable chemical compositions of TQBQ-COF.

**Morphology characterizations.** The scanning electron microscope (SEM) images (Fig. 2a, Supplementary Fig. 8a, b) show a porous-honeycomb morphology of TQBQ-COF with irregular pores, which can also be observed from the TEM image (Supplementary Fig. 8c). As TEM result has shown the d-spacing of ~3.0 Å between each TQBQ-COF monolayer, a ~5.0 nm height of TQBQ-COF layers is corresponding to 16 layers of TQBQ-COF monolayers in the AFM image (Fig. 2b), which is smaller than that of graphene sheets (0.34 nm). In addition, Raman spectrum (Supplementary Fig. 8d) gives two peaks at ~1326 cm⁻¹ (D bands) and 1521 cm⁻¹ (G bands), where the D bands are derived from the sp³ C and bent sp² C structures and the G bands referring to the conjugated sp² C structure. Combined with AFM and TEM results, the value of $I_D/I_G$ (0.84) from Raman spectrum suggests moderate defect level for the 2D structure of TQBQ-COF layers[35].

N₂ adsorption-desorption isotherms measurements were performed to test the surface area and pore distribution of TQBQ-COF powder. The pristine TQBQ-COF with a specific surface area of 46.95 m² g⁻¹ was shown in Supplementary Fig. 9. After annealing it at 200 °C for 5 h, the specific surface areas of the heat-treated sample can reach up to 94.36 m² g⁻¹, with a micropore size of 1.18 nm (Supplementary Fig. 10). The enlarged surface area can be attributed to the volatilization of small molecules absorbed into the holes. According to the isotherm type (Fig. 2c), most of void spaces in TQBQ-COF come from mesopores. The t-Plot analyses (the inset of Fig. 2c) was added to elucidate the microporous properties of TQBQ-COF. The t-Plot results show that the micropores exist inside the TQBQ-COF. Therefore, the most pores of TQBQ-COF are mesopores, accompanied with some micropores.

Thermogravimetric analysis of the as-prepared TQBQ-COF shows a weight loss below 120 °C due to the small molecules absorbed into the mesopores, while the heat-treated sample

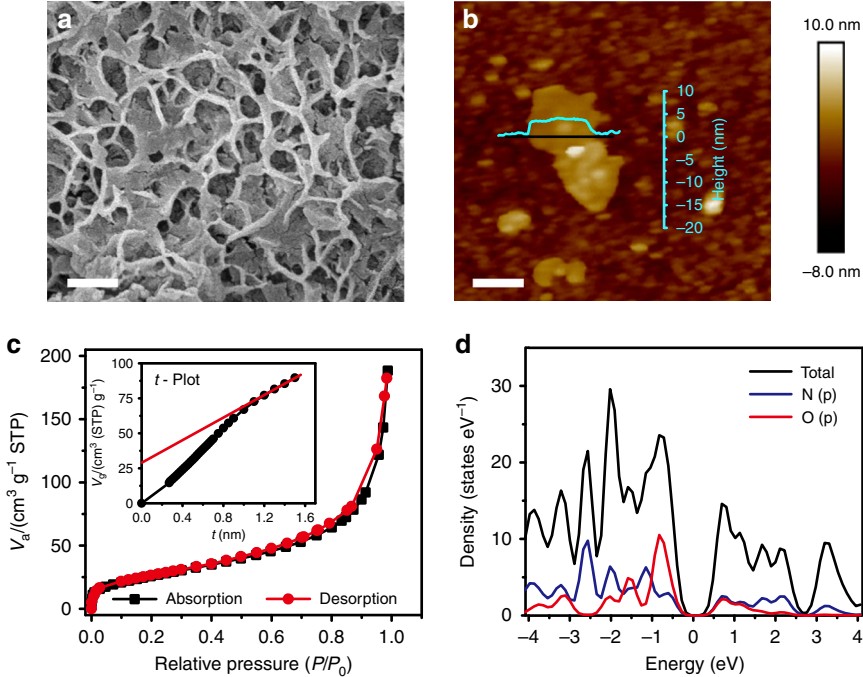

**Fig. 2 Characterizations of TQBQ-COF. a** The SEM image of TQBQ-COF powder. Scale bar, 100 nm. **b** AFM height image of TQBQ-COF material. The inserted line shows a height of ~5.0 nm for 16 monolayers of TQBQ-COF. Scale bar, 300 nm. **c** $N_2$ adsorption/desorption isotherms of the heat-treated TQBQ-COF. The inset is the $t$-Plot analyses of the heat-treated sample. **d** The DOS of the simulated AB stacking structure for TQBQ-COF.

displays higher thermal stability (Supplementary Fig. 11). The heat-treated TQBQ-COF with larger surface area and more stable structure can favor ion diffusion and accommodate the intercalation of large $Na^+$ ions. Furthermore, the electronic conductivity is achieved as $1.973 \times 10^{-9}\,S\,cm^{-1}$ according to the linear sweep voltammetry curves (Supplementary Fig. 12). The calculated band gap of TQBQ-COF is less than 1.0 eV according to the density of states (DOS) (Fig. 2d), demonstrating the intrinsic semiconductor property of TQBQ-COF. Note that the ionic conductivity of as-prepared TQBQ-COF was not tested because it does not contain $Na^+$. Instead, we tested the ionic conductivity of the discharged product (discharged to 0.8 V) by electrochemical impedance frequency response[36]. As shown in Supplementary Fig. 13, the high-frequency part relates to the electronic and ionic resistance ($R_e$ and $R_i$), and the low-frequency part corresponds to the electronic resistance ($R_e$). Therefore, the ionic conductivity is calculated to be as high as $5.53 \times 10^{-4}\,S\,cm^{-1}$ (detailed calculation process can be seen in the Supplementary Methods). The conductive TQBQ-COF material with rich nitrogen atoms and porous conjugated structure is able to facilitate electrons transport and $Na^+$ ions diffusion among the abundant redox sites[27].

**In/Ex-situ FTIR and ex-situ XPS on the TQBQ-COF electrodes**. Before the study of the sodium-storage mechanism, the electrochemical performance of the TQBQ-COF electrode is investigated through the assembled coin-type SIB. The electrochemical impedance spectroscopies (EIS) of the TQBQ-COF electrodes were firstly tested in the electrolyte of 1.0 M $NaPF_6$ dissolved in diethylene glycol dimethyl ether (DEGDME) and propylene carbonate (PC), respectively (Supplementary Fig. 14). The charge transfer resistance (semicircles in high-frequency regions) of TQBQ-COF electrode in $NaPF_6$/DEGDME (150 Ω) is much less than that in $NaPF_6$/PC (700 Ω). This can be attributed to the lager radius of $[NaPC3]^+$ solvation configuration and the high charge transfer resistance of Na anode in the PC-based electrolyte (Supplementary Figs. 15, 16, and 17). Furthermore,

TQBQ-COF is indeed insoluble in $NaPF_6$/DEGDME (Supplementary Fig. 18). However, the electrolyte of $NaPF_6$/DEGDME can wet the TQBQ-COF electrode plate in 5 s (Supplementary Fig. 19), implying the good wettability of $NaPF_6$/DEGDME for the TQBQ-COF electrodes. Therefore, the applied electrolyte is 1.0 M $NaPF_6$/DEGDME in the following experiments.

In-situ FTIR was applied to investigate the sodium storage mechanism of the TQBQ-COF electrode during electrochemical process. Figure 3a shows the discharging and charging curves (0.8–3.7 V, at 0.02 A $g^{-1}$) of TQBQ-COF electrode for the initial two cycles. Notably, the TQBQ-COF electrode can achieve a high capacity of 505.3 mAh $g^{-1}$ (corresponding to 12 $Na^+$ ions with each TQBQ-COF repetitive unit) when discharged the cell to 0.8 V, and exhibit a reversible performance when recharged to 3.7 V. The characteristic peaks of carbonyls and pyrazines for pristine TQBQ-COF electrode are located at ~1627 and 1545 $cm^{-1}$, respectively. As shown in Fig. 3b, the two characteristic peaks gradually become weak in the discharging process, coinciding with the sequential coordination of $Na^+$ ions with the active sites on the TQBQ-COF layer. The peak at 1627 $cm^{-1}$ almost disappears when discharged to 0.8 V, while the characteristic peak at 1545 $cm^{-1}$ is still leaving a faint peak. Combined with the ex-situ FTIR results (Supplementary Figs. 20 and 21), the peak of the C=C mode could be assign to ~1600 $cm^{-1}$. Thus, the faint peak (1550–1620 $cm^{-1}$) at fully discharged state in the ex-situ FTIR results is mostly owing to the existing of C=C and the residual C=N for $Na_nTQBQ$-COF ($n = 1$–12). It is noted that the in-situ FTIR is helpful to detect the sequential structure change for TQBQ-COF electrode, but it could not eliminate the interference of the electrolyte and the testing condition. The in-situ and ex-situ FTIR measurements could work well with each other to exhibit the $Na^+$ storage process. The $Na^+$ storage ability for both carbonyls and pyrazines of the TQBQ-COF electrode can be confirmed by the corresponding peaks change. In the charging process, the characteristic peaks of C=O and C=N reemerge gradually and ultimately strengthen back to the pristine state. The same phenomenon

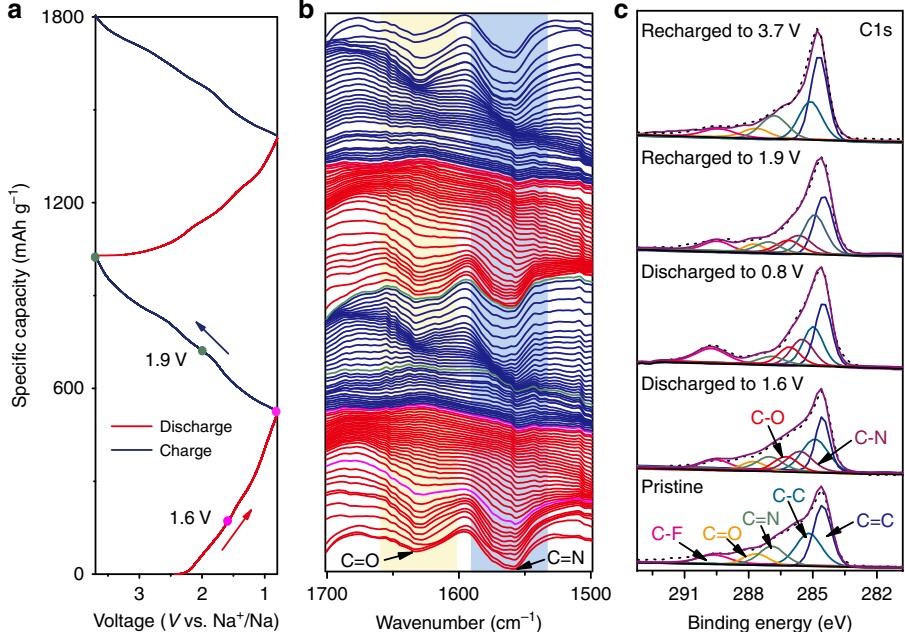

**Fig. 3 Sequential structure evolution of active sites for TQBQ-COF electrodes during sodiation/desodiation processes. a** Discharge and charge profiles of TQBQ-COF electrode at 0.02 A g$^{-1}$ in the voltage range of 0.8–3.7 V in the initial two cycles. **b** In-situ FTIR spectra collected at different states corresponding to **a**. **c** The C1s XPS spectra of TQBQ-COF electrodes at different charge/discharge states marked in **a**.

occured in the second cycle, showing a reversible intercalation/deintercalation process of Na$^+$ in the TQBQ-COF electrode.

The sodiation/desodiation process was further investigated by the ex-situ XPS spectra in Fig. 3c. The peaks of C = O (287.6 eV) and C=N (287.0 eV) weaken gradually when discharged the cell to 1.6 V, and become very weak as discharged to 0.8 V. In contrast, the peaks of C–N (285.6 eV) and C–O (286.2 eV) emerge gradually during the discharging process, and show obvious when discharged to 0.8 V. The evolution of peaks of C = O, C=N, C–O, and C–N all exhibit a reverse trend, and all peaks in the C1s spectra are almost identical to the pristine state when the battery is recharged to 3.7 V. The results indicate that both carbonyl groups and pyrazine sites have involved in the total sodiation and desodiation processes, which is in good agreement with the in/ex-situ FTIR results. The in/ex-situ FTIR spectra manifest that the carbonyl groups and pyrazine sites can almost simultaneously react with Na$^+$ ions. Moreover, the ex-situ XPS and the following DFT calculations could support this result as well. The possible reason why O and N atoms simultaneously react with Na$^+$ ions is the minor difference in electronegativity between O and N atoms on the conjugated structure of TQBQ-COF.

**DFT calculations during sodiation/desodiation**. DFT calculations were applied to identify the sequential structural evolution of TQBQ-COF during sodiation and desodiation processes. Firstly, the DFT calculations were used to calculate the optimized structures of TQBQ-COF layers and the configurations after different degrees of sodiation. The molecular structure of TQBQ-COF is simulated by one ring unit, whose outer edges are saturated with hydrogen atoms, as shown in Fig. 4a. After optimizing, the configuration of one TQBQ-COF unit is determined to be a planar hexagonal structure with a micropore diameter of 11.4 Å, which is in good agreement with the BET results (with a micropore diameter of 1.18 nm, Supplementary Fig. 10). Secondly, using molecular electrostatic potential (MESP) method[37], six equivalent minima of ESP value are found in the middle of

two adjacent nitrogen atoms within the molecular plane. These ESP minima correspond to six Na$^+$ accommodation sites. Thus, 6 Na$^+$ ions are placed at these sites and the consequent Na$_6$TQBQ-COF unit is optimized to be a slightly curled surface, which can be attributed to the tension among both nitrogen and oxygen atoms and sodium ions.

To clarify the stabilization effect of both nitrogen and oxygen atoms towards Na$^+$, orbital composition analyses of the LUMO of TQBQ-COF molecule was performed (Fig. 4b). The results show that nitrogen atoms account for 22.21% of the LUMO composition while that of oxygen atoms is 19.72%. The similar percentage of nitrogen and oxygen in the orbital composition of LUMO indicates that both nitrogen and oxygen atoms stabilized the Na$^+$ ions collaboratively, which are in good accordance with the experimental FTIR and XPS tests where the intensities of peaks of C=N bonds and C=O groups weaken simultaneously. In addition, the MESP and LUMO patterns of the two TQBQ-COF repetitive units (Supplementary Fig. 22) also exhibit same priority of C=O in accommodating Na$^+$ ions, confirming that one repetitive ring unit is sufficient for simulating all C = O groups in an extended structure of TQBQ-COF. As a result, the first 6 Na$^+$ ions storage process should be depicted as per Na$^+$ accommodated between every two adjacent nitrogen atoms as well as two oxygen atoms in per unit plane of TQBQ-COF.

The MESP of Na$_6$TQBQ-COF is calculated to find the rest of the Na$^+$ ion accommodation sites. A total of 12 ESP minima are found on a Na$_6$TQBQ-COF unit, in which each carbonyl oxygens accommodated 2 ESP minima with a 1.8-Å distance between the minimum and the molecular surface. To prevent an over crowded sodiated configuration which would lead to large steric hindrance, one Na$^+$ is placed near each carbonyl in Na$_6$TQBQ-COF unit to form the initial structure of Na$_{12}$TQBQ-COF. The optimized geometry of Na$_{12}$TQBQ-COF is also a curled surface similar to that of a Na$_6$TQBQ-COF unit. Moreover, the accurate atomic coordinates of Na$_{12}$TQBQ-COF are provided in Supplementary Table 2. From the atomic coordinates of Na$_{12}$TQBQ-COF, we can obtain the Mulliken charges of each atom (such as O, N, Na). The Mulliken charges of two Na atoms with different chemical

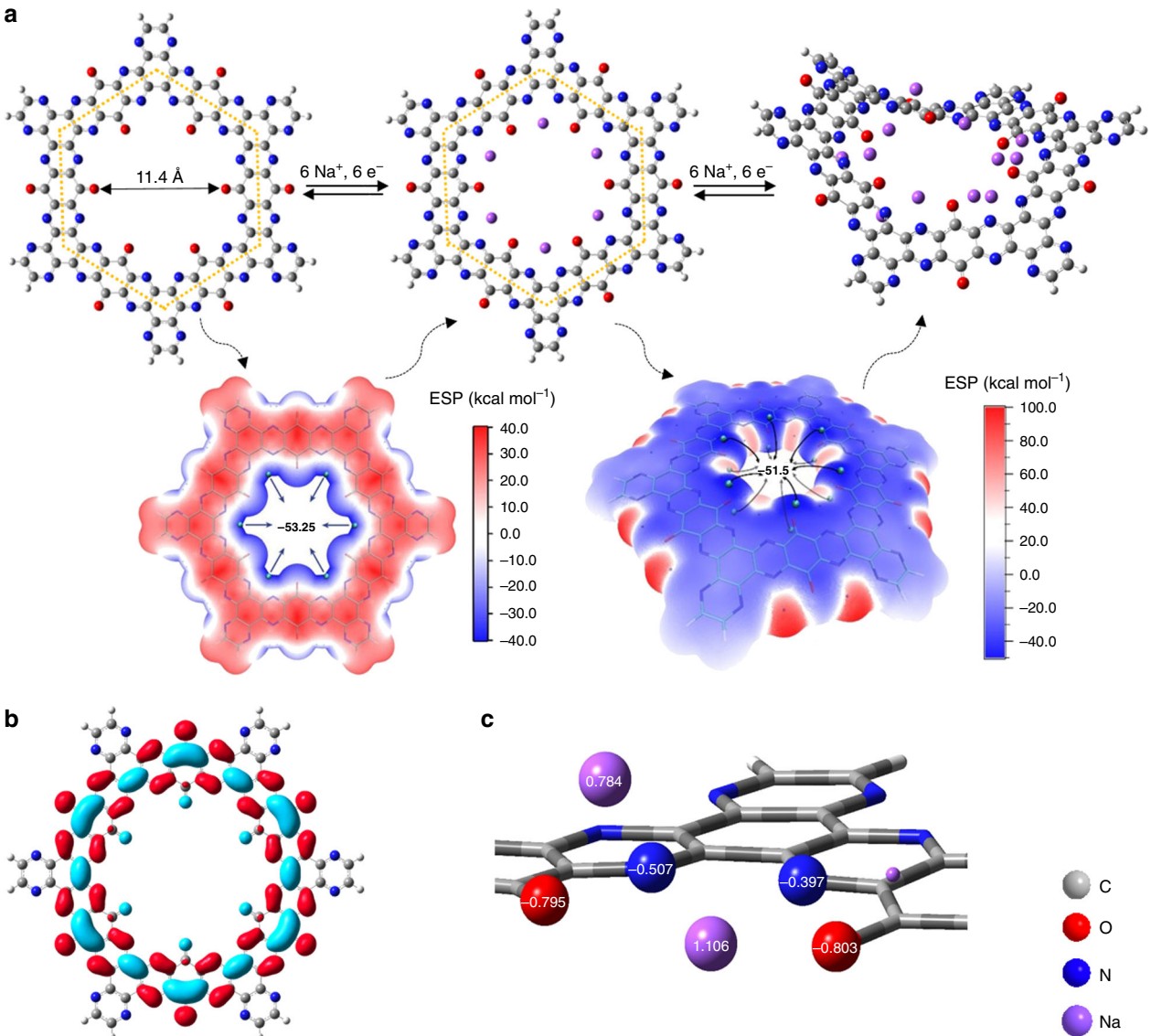

**Fig. 4 The up-to 12 Na$^+$ ions redox chemistry mechanism for each TQBQ-COF repetitive unit. a** Schematic diagram of two-step sodiation and desodiation process of the TQBQ-COF electrode obtained via the molecular electrostatic potential (MESP) method. The diameter of the TQBQ-COF unit by the DFT calculations is 11.4 Å, and the values inside the ring unit of the MESP figures indicate the ESP minima. **b** LUMO plot for per TQBQ-COF repetitive unit. **c** Mulliken charges of two Na atoms with different chemical environment and their adjacent N and O atoms in Na$_{12}$TQBQ-COF.

environment and their adjacent N and O atoms are marked in Fig. 4c. Furthermore, the average atomic valences of Na, O, and N in Na$_{12}$TQBQ-COF are calculated to be +1, −1, and −0.5, respectively. The coordination environment of the two sets of Na$^+$ ions in Na$_{12}$TQBQ-COF is also analyzed using Mayer bond order in Supplementary Fig. 23 and Supplementary Table 3. The sodium ions located within the molecular plane of the TQBQ-COF are coordinated by all four nitrogen and oxygen atoms (O-29, O-123, N-30, and N-56), while the sodium ions outside the molecular plane are only coordinated by one nitrogen atom (N-56) and one oxygen atom (O-123). Further MESP calculation on Na$_{12}$TQBQ-COF exhibits no Na$^+$ accommodation site in the unit ring (Supplementary Fig. 24), confirming that only 12 Na$^+$ ions can access to each unit of the TQBQ-COF. Therefore, the next 6 Na$^+$ ions storage process can be depicted as per Na$^+$ located between two adjacent oxygen and nitrogen atoms outside of TQBQ-COF plane.

To further understand the sequential energy evolution of the TQBQ-COF accompanying the multiple-electron transfer, the corresponding voltages during sodiation are also calculated by DFT calculations. The energy values of Na$_n$TQBQ-COF ($n$ = 1, 2, 5, 6, 7, 11, and 12) are defined by single point energy and Gibbs free energy (Supplementary Table 4). Thus, their corresponding redox potentials can be obtained (Supplementary Fig. 25). The average voltage plateaus corresponding to the first 6 Na$^+$ ions and following 6 Na$^+$ ions storage are calculated to be 2.79 V and 1.59 V, respectively.

**Electrochemical performance.** Subsequently, we investigated the electrochemical performance of TQBQ-COF electrode. The CV curves display two obvious couples of redox peaks (Fig. 5a), which can be ascribed to the successive two-step sodiation/desodiation process. The peak at 2.2 V is in consistent with first 6 Na$^+$ ions located inside the ring plane of TQBQ-COF, while the peak at 1.5 V can be assigned to the next 6 Na$^+$ ions stored outside the ring plane of TQBQ-COF, which are in good agreement with DFT calculations. The galvanostatic charge–discharge

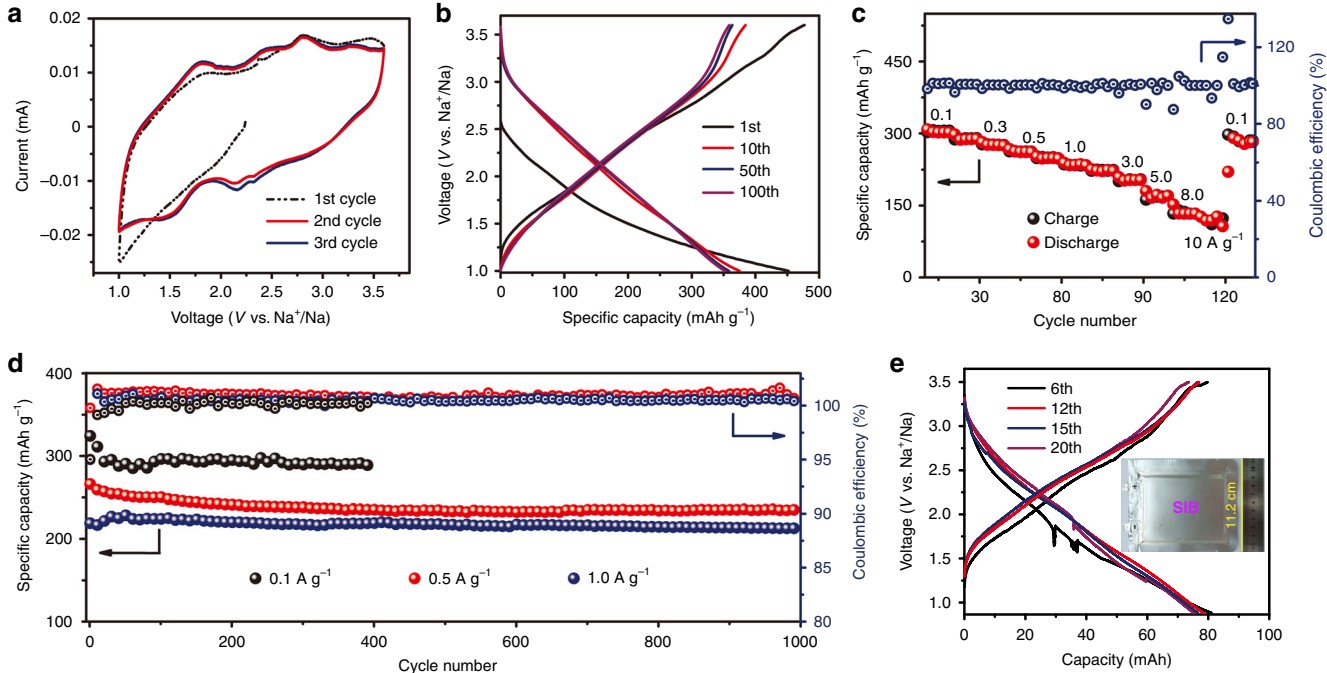

**Fig. 5 Electrochemical properties of the TQBQ-COF electrodes between the voltage range from 1.0 V to 3.6 V. a** CV curves of the TQBQ-COF electrode at a scan rate of 0.2 mV s$^{-1}$. **b** Charge–discharge profiles of the TQBQ-COF electrode at 0.02 A g$^{-1}$. **c** The rate performance of the TQBQ-COF electrode from the current density of 0.1 to 10 A g$^{-1}$, then back to 0.1 A g$^{-1}$. **d** Long cycling stability of TQBQ-COF electrodes at different current densities (0.1, 0.5, and 1.0 A g$^{-1}$). The capacities are all calculated based on the mass of TQBQ-COF. **e** The selected charge/discharge curves of pouch-type Na//TQBQ-COF batteries at a current of 50 mA. The capacity is calculated based on the mass of the full cell.

profile of the TQBQ-COF electrode in sodium battery delivers an initial discharge capacity of 452.0 mAh g$^{-1}$ (at 0.02 A g$^{-1}$, 1~3.6 V) in Fig. 5b. It exhibits a high reversible capacity of ~400 mAh g$^{-1}$ in the following 10 cycles and maintains a capacity of 352.3 mAh g$^{-1}$ after 100 cycles. In addition, the compaction density of the TQBQ-COF pellet calculated from Supplementary Fig. 26 is ~1.63 g cm$^{-3}$, thus the volumetric energy of the TQBQ-COF electrode is calculated to be ~1252 Wh L$^{-1}$$_{TQBQ-COF}$ (based on the capacity of 452.0 mAh g$^{-1}$$_{TQBQ-COF}$ at 0.02 A g$^{-1}$). The wide scan of XPS spectra (Supplementary Fig. 27) manifest some new peaks in the TQBQ-COF electrode after recharged to 3.7 V, which is associated with the formation of solid electrolyte interface (SEI) around the TQBQ-COF cathode. The capacity loss could be explained by the XPS results (formation of SEI) or some unstable sites for Na$^+$ storage. As shown in Supplementary Fig. 28, the self-discharge performance was evaluated after standing the fully-charged cell (after 5th cycle) for 24 h, and 92% of original capacity was retained after fully discharged at 0.2 C (1 C = 400 mAh g$^{-1}$). The good sodium storage performance could be attributed to the stability of the TQBQ-COF material and less side reactions of the electrolyte.

The TQBQ-COF electrode shows reversible capacities of 278.6, 234.0, 180.6, and 134.3 mAh g$^{-1}$ (Supplementary Fig. 29) at 0.3, 1.0, 5.0, and 10.0 A g$^{-1}$, respectively. As the current density was decreased to 0.1 A g$^{-1}$, the capacity of TQBQ-COF electrode could also reach up to ~300 mAh g$^{-1}$. Figure 5c presents the corresponding rate performance of the TQBQ-COF electrode. The coulombic efficiency varies once the current density changes, which is likely caused by the concentration polarization inside the battery. Notably, an impressive capacity of 134.3 mAh g$^{-1}$ can be achieved after discharging the cell in only 48 s (at 10.0 A g$^{-1}$), showing an excellent rate performance. CV measurement was employed to estimate diffusion kinetics of

Na$^+$ at different scan rates, ranging from 0.2 to 2.0 mV s$^{-1}$ (Supplementary Fig. 30). Based on the equation:

$$i = av^b \qquad (1)$$

where $i$ is the peak current, $v$ is the scan rate, and $a$ and $b$ are constants. When the value of $b$ is close to 0.5, it shows a Na$^+$ diffusion controlled process; while it presents a supercapacitor behavior when $b \approx 1$. The battery with TQBQ-COF electrode shows the value of $b$ is close to 0.9, which is in consistent with the supercapacitor-like rate performance, delivering fast kinetics for Na$^+$ storage[34]. The Na$^+$ diffusion coefficient ($D_{Na}^+$) was calculated to be ~10$^{-11}$ S cm$^{-1}$, both by the CV methods[38] and the galvanostatic intermittent titration technique measurement (Supplementary Fig. 31)[29]. The detailed calculation process for the $D_{Na}^+$ can be seen in the Supplementary Methods. As a result, the good reaction kinetics, high porous feature and nitrogen-doping conjugated structure guarantee the TQBQ-COF electrodes with high capacity and high rate capability for sodium batteries.

The specific capacity of TQBQ-COF electrode can reach 327.2 mAh g$^{-1}$ at 0.1 A g$^{-1}$ in Fig. 5d, and a capacity retention of ~89% after 400 cycles is achieved. Moreover, the long cycling stability of the TQBQ-COF electrode is obtained (Fig. 5d). After 1000 cycles at 0.5 A g$^{-1}$ and 1.0 A g$^{-1}$, they exhibit a reversible capacity of 236.5 and 213.6 mAh g$^{-1}$, yielding a capacity retention of 91.3% and 96.4%, respectively. The specific capacity of the TQBQ-COF electrode shows an increasing trend for the initial 10 cycles at the large currents owing to the slow process to open the ion transport channel for the TQBQ-COF electrode, and the capacity retention is calculated based on the 10th cycle. Moreover, the TQBQ-COF delivers a high Coulombic efficiency of ~100%, and the energy efficiency is around 85%

(Supplementary Fig. 32), showing good cycling stability and high-energy efficiency[39].

The excellent cycling stability of the TQBQ-COF electrode could be ascribed to its inherent stable feature and the high $D_{Na^+}$. Additionally, the weak peak in the UV-vis spectra of TQBQ-COF electrodes in $NaPF_6$/DEGDME indicates the poor solubility of the discharged and recharged products (Supplementary Fig. 33). TQBQ-COF electrode remains low impedance after 500 (169.4 Ω) and 1000 cycles (220.2 Ω) in Supplementary Fig. 34, and shows no obvious cracks in the plate after 500 cycles (Supplementary Fig. 35). The TQBQ-COF electrode displays high-energy density and upper power density (Supplementary Fig. 36 and Supplementary Table 5) when compared with current reported organic cathodes for sodium batteries. Furthermore, a pouch-type 81 mAh sodium battery with Na anode, TQBQ-COF cathode, and $NaPF_6$-DEGDME electrolyte was constructed, showing a capacity of 75.3 mAh after 20 cycles at a current of 50 mA (Fig. 5e). The detailed assembly process of the pouch-type sodium batteries can be seen in Supplementary Methods. The capacity (54.4 mAh $g^{-1}_{cell}$) for the 6th cycle normalized by the whole mass (1.49 g) of the pouch-cell is shown in Supplementary Fig. 37 and Supplementary Table 6. With a voltage plateau of 1.86 V, the pouch-type sodium battery exhibits an outstanding energy density of 101.1 Wh $kg^{-1}$ (based on the mass of the whole cell), demonstrating the potentially practical applications of TQBQ-COF. As the volume of the pouch-cell is 1.92 ($6 \times 8 \times 0.04$) $cm^3$, the volumetric energy of the total pouch-cell is calculated to be 78.5 Wh $L^{-1}$. The porous and nitrogen-doped structure of TQBQ-COF facilitates the rapid ions/electrons transport among the multiple active sites, and hence endows the TQBQ-COF electrodes with outstanding cycling stability and rate performance for sodium batteries.

## Discussion

We have successfully synthesized a TQBQ-COF with multiple active sites and nanoporous structure for high-efficiency electrochemical sodium storage. The pyrazines as well as carbonyls act as redox sites in the nitrogen-rich conjugated structure of TQBQ-COF. The up-to 12 $Na^+$ redox chemistry mechanism of each TQBQ-COF repetitive unit is demonstrated by the in/ex-situ FTIR methods and DFT calculations. The insoluble TQBQ-COF electrode exhibits a remarkably high specific capacity of 452.0 mAh $g^{-1}$ at 0.02 A $g^{-1}$, excellent cycling stability (~96% after 1000 cycles at 1.0 A $g^{-1}$) and good rate capability (134.3 mAh $g^{-1}$ at 10.0 A $g^{-1}$). The superior rate performance can be attributed to the honeycomb-like morphology and high electronic conductivity (1.973 × $10^{-9}$ S $cm^{-1}$) of TQBQ-COF. Moreover, the TQBQ-COF-based pouch-type battery shows a capacity of 81 mAh and a voltage plateau of 1.86 V, corresponding to the gravimetric and volumetric energy density of 101.1 Wh $kg^{-1}$ and 78.5 Wh $L^{-1}$ (based on the whole pouch-type cell), respectively. This work sheds light on the design and application of COFs with multiple redox sites for high-energy and high-power sodium batteries.

## Methods

**Synthesis of TQBQ-COF.** CHHO (5.0 g, 16.0 mmol) and TABQ (4.0 g, 24.0 mmol) were put into the flask bottom under argon atmosphere, and 150 mL mixture of deoxygenated acetic acid/ethanol (v/v 1:1) was slowly added. After stirred at room temperature for 30 min, the mixture was heated at 100 °C for 48 h, followed by heated at 140 °C under Ar bubbling for another 6 h. A dark-red powder was achieved after rinsed by massive water, methanol, and acetone, respectively. Moreover, the resultant powder was further washed with water and methanol by Soxhlet extractor for 12 h. The sample was finally annealed at 200 °C for 5 h under argon atmosphere.

**Material characterization.** The structure of the resultant TQBQ-COF material was examined by solid-state $^{13}C$ NMR with Inova 400 MHz Spectrometer (Varian Inc., USA), and Fourier transform infrared spectroscopy (FTIR, Bruker

5700 TENSOR II) in range of 400–4000 $cm^{-1}$. Powder XRD (Rigaku Mini-Flex600 × -ray generator, Cu Kα radiation, $\lambda = 1.54178$ Å), high-resolution transmission electron microscopy (HRTEM) and the selected area electron diffraction (SAED) pattern (Taols F200X G2) were applied to investigate the crystallinity and the microstructure of the TQBQ-COF powder. The elemental distributions of the TQBQ-COF material and the relevant electrodes before and after discharge/charge were characterized by scanning electron microscopy-Energy dispersive spectrum mapping (SEM-EDS), elemental analysis (EA, vario EL CUBE), and X-ray photoelectron spectroscopy (XPS, Perkin Elmer PHI 1600 ESCA), respectively. The morphologies of the TQBQ-COF material and the relevant electrodes were observed by scanning electron microscopy (SEM, JEOL JSM7500F), transmission electron microscopy (TEM, Taols F200X G2), and $N_2$ adsorption/desorption measurement (BEL Sorp mini). Moreover, Raman (DXR Microscope, Thermo Fisher Scientific with excitation at 532 nm) and TG-DSC analyzer (NETZSCH, STA 449 F3) were separately carried out to examine the structure and the stability of TQBQ-COF material, respectively.

**TQBQ-COF.** $^{13}C$ NMR (400 MHz, Magic Angle Spinning): δ (ppm): 210 (m); 172 (m); and 145 (m). Elemental analysis: calculated for $C_{30}N_{12}O_6$: C, 57.69%; N, 26.92%; O, 15.38%; found: C, 53.12%; N, 23.76%; H, 2.17%. FTIR (ATR, $cm^{-1}$): 1,627; 1,545; 1,367; 1,263; 1,098; 1,022; and 802.

**Electrochemical measurements.** The electrochemical performance of the TQBQ-COF electrodes including galvanostatic charge/discharge, CV and EIS were evaluated in CR2032-type coin cells with sodium disks applied as the counter electrodes. Moreover, the glass microfiber membrane (Whatman GF/D, Aldrich) was used as the separator, and 1.0 M $NaPF_6$/DEGDME solution was applied as the electrolyte. The TQBQ-COF electrode was prepared by mixing 50 wt% TQBQ-COF powders, 40 wt% Super P and 10 wt% polyvinylidene fluoride (PVDF) in the homogenate locket, dispersing the mixture in anhydrous N-methyl-2-pyrrolidinone (NMP) and casting the resulting slurry on an Al foil, followed by drying it at 80 °C in vacuum for 12 h. Finally, punched the Al foil into circular electrodes and stored them in Ar-filled glovebox before the assembly of the cells. Galvanostatic charge/discharge was performed on LAND-CT2001A battery instrument (LAND Electronic Co., Wuhan China). CV was carried out in the voltage range of 0.8~3.8 V at a scan rate of 0.2 mV $s^{-1}$ and EIS was carried out on a Parstat 263 A electrochemical workstation (AMETEK Co.) in a frequency range of $10^5$–0.01 Hz. With a cathode-limited design, the cell capacity was determined based on the mass of TQBQ-COF by deducting the capacity contribution from Super P.

**In/Ex-situ FTIR and ex-situ XPS spectroscopy.** To monitor the structural evolution of active materials in charge/discharge processes, the TQBQ-COF electrodes for in-situ FTIR measurement were containing 80 wt% active material, 10 wt% Super P, and 10 wt% PTFE. After hand mixing, the slurry was casted onto the stainless steel net, and the electrodes were dried at 60 °C for 12 h under vacuum. The process to prepare the cell for in-situ FTIR testing is in the Ar-filled glove box. During the test, a stream of Ar flow was used to protect the discharging/charging products from oxidized in attenuated total reflection (ATR) pattern. Based on the CV results, the assembled cells were cycled at a current density of 0.02 A $g^{-1}$ in the range of 0.8–3.7 V for in-situ FTIR measurements. TQBQ-COF electrodes for ex-situ FTIR testing contain 70 wt% active material, 20 wt% Super P and 10 wt% PVDF. The preparation was made in the same way as the approach mentioned above. The samples for ex-situ FTIR tests were obtained by disassembling the labeled cells in the argon-filled glovebox, and washing the electrodes with glycol dimethyl ether (DME) for three times, followed by dried in vacuum. The products at different charge and discharge states were tested in a stream of Ar flow using the ATR pattern.

TQBQ-COF electrodes for XPS testing contain 60 wt% active material, 30 wt% Super P, and 10 wt% PVDF. Samples were prepared by disassembling the labeled cells in the Ar-filled glovebox, and washing the electrodes with glycol dimethyl ether (DME) for three times, followed by dried in vacuum. Finally, the products at different charge and discharge states were tested in the argon atmosphere.

**Density functional theory calculations.** Becky's three-parameter exchange function combined with Lee-Yang-Parr correlation functional (B3LYP) method with 6–31 G (d) basis set[40] was applied for the geometric optimization of the TQBQ-COF. All molecular structures were optimized using Gaussian 16 software package under B3LYP/6–31 G (d) level of theory followed by vibrational frequency calculations and to further confirm their stability. To simulate the solvation effect, single point energy calculations were performed using the SMD solvation model at B3LYP/6-31 + G (d, p) level with a solvent dielectric constant (ε)[41] of 7.2 which reliably describes the polarity of experimentally used electrolyte solvent DEGDME. The molecular electrostatic potential (MESP) method was applied to predict the sodiation sites of TQBQ-COF using Multiwfn 3.6 software[42], by which the Mayer bond order analysis[43] and the density of states (DOS) were also carried out. In addition, the accurate atomic coordinates of $Na_{12}$TQBQ-COF were calculated by Gaussian 16 software package, from which the Mulliken charges of two types of Na atoms and their adjacent N and O atoms can be obtained.

The sodiation/desodiation process of the potential active sites was revealed by orbital composition analyses of the lowest unoccupied molecular orbital (LUMO), to clarify the stabilization effect of both nitrogen and oxygen atoms towards sodium ions. The MESP of TQBQ-COF and $Na_6$TQBQ-COF were calculated to find the rest of $Na^+$ accommodation sites. In addition, the average voltages of the above two stages were predicted by the equation:

$$V = -\frac{\Delta G}{nF} \qquad (2)$$

where $V$ is the voltage, $\Delta G$ is the standard Gibbs free energy change during the two sodiation processes, $n$ is the number of electrons transferred, and $F$ is the Faraday constant.

## Data availability

The data that support the findings of this study are available from the corresponding author upon reasonable request.

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

## Acknowledgements

This work was supported by the National Programs for Nano-Key Project (2017YFA0206700), the National Natural Science Foundation of China (21835004), and 111 Project from the Ministry of Education of China (B12015). The calculations in this work were performed on TianHe-1(A), National Supercomputer Center in Tianjin.

## Author contributions

J.C. proposed the concept and supervised the work; R.J.S. designed and performed the experiments; L.J.L. performed the DFT calculations and explained the sodiation process; Y.L. and C.C.W. performed the batteries assembly; Z.H.Y., Y.X.L. and L.L. helped to perform the experiments and analyze the data; all authors contributed to the discussion and the manuscript preparation.

## Competing interests

The authors declare no competing interests.
