## [Peer Review File · Nature Communications]

Reviewers' comments:

Reviewer #1 (Remarks to the Author):

This manuscript reports an organic framework cathode for sodium-ion batteries. This new cathode shows a high gravimetric energy density of 463.2 mAh/g. The contents are of great interests however the manuscript does not sufficiently provide data enough to convince advantages of their newly synthesized cathode material compared to present oxide-based electrode materials. Therefore, this referee requests the authors to show following data and discussions in order to assess further whether the content of manuscript can be recommended to publish in this journal.

1. Characterizations are not enough. First of all, please bring Fig. S4 in the main manuscript. This data is very important. This referee believes that the authors should be improve this data. First on Figure S4a, there are a lot of mismatching between the simulation and the experimental pattern. The authors should show residual and discuss about the reason for difference in their model and the prepared framework. Furthermore, Figure S4d is quite unclear data. There is no scale, no electron beam direction and no indication which point is corresponding (002).
2. Please measure the conductivity of the material. The particle size is quite large (ca 1 micro meter?) therefore conductivity plays a key role for electrochemical properties.
3. Why graphene's D and G bands are same for this organic framework's bands? There is a robust theory to calculate the position of G band, and from this point this referee are not convinced why this material shows its G band at an identical position to graphene.
4. The referee sees that the electrochemical properties section should be improved. The range to show Coulombic efficiency (%) should be from 80 to 100 %. Nobody can check how CE changing on increasing cycling number. In case of organic framework, an important point is energy efficiency (i.e. Q at discharge/Q at charge). Therefore, please show the energy efficiency. Furthermore, how is the volumetric energy density? It is much more important value than gravimetric energy density.
5. About pouch cell: please show the capacity normalized by the whole mass of cell. With this, please show the amount and mass of electrolyte and sodium metal.

Reviewer #2 (Remarks to the Author):

Chen et al. reported COFs-based cathode materials with a high theoretical capacity for sodium-ion batteries. This work represents the first report combining N-heteroaromatic triquinoxalinyne and benzoquinone in a single material. Synthesis, structural characterization, morphology characterization, In-situ FTIR, ex-situ XPS, DFT calculation, battery performance, were carried out. However, some issues in the work should be addressed before I could make a reasonable evaluation.

1. In Figure 1a, the author seems to be providing a unit with a $18e-12$ Na⁺ reduction of the inner shell. Could the author provide an accurate structure, specifically on the bond valence?
2. In Figure S17, the structure is not a repeating unit, and therefore the calculation of theoretical capacity (515 mAh/g) is wrong because of this wrong drawing. The last four structures of reduced products are not in reduced states.
3. In the FTIR spectra, can the authors assign the vibration mode of C=C? This mode seems to be relatively strong in the TABQ.
4. The in-situ FTIR curves is too crowded, and the range is too narrow. Could the author put some important curves in one supporting figure? Can we know the reduction sequence for C=O and C=N according to the in-situ FTIR result?
5. In Figure S3, the D band in Raman spectrum may not only originated from the "defection of the graphene-like structure". It could be caused by the sp³ C and bent sp² C structures. And such results

do not provide implications for a “2D layer structure”. AFM and TEM (Figure S4 and S5) could provide more insights.

6. Is the XPS data/peak position corrected with a standard component? Although TQBQ-COF is a n-type semiconductor, the peak positions may still suffer from deviation from correct positions.

7. The XPS fitting results in Figure S7 show a wide distribution of FWHM. In Figure S7, the C 1s BE of C=N is lower than that of C=C. The author stated that the lower BE may be caused by the delocalization of lone pair electrons of N over the conjugated structure. Could the author give more solid evidence, since such low BE in C 1s is rarely observed, even in g-C₃N₄?

8. The C=O BE in Figure S7 is high as ~288 eV, which is not usually seen in conjugated C=O moieties. If as the author stated that N is delocalizing electron over the system, how could the BE of C 1s in C=O is even higher than the conventional observed values (~287 eV). What is the possible effect of N on the XPS of C 1s spectrum? In Figure S7b, the label C=N appeared twice, and that for C-N is missing.

9. The authors found that the as-prepared TQBQ-COF may have absorbed small molecules as evidenced by the TGA result. In my opinion, the FTIR data also suggests this hypothesis. The C-H mode at ~2900 cm⁻¹ diminished significantly after heat treatment.

10. What is the possible reason for the high charge transfer resistance in the PC-based electrolyte? Is it due to the solvation of Na⁺ or the interaction of solvent molecules with the TQBQ-COF backbone?

11. In Figure 2, what is the mechanism of “the N-fused conjugated skeleton” to induce existence of a peak component at 1545 cm⁻¹ at a fully discharged state? Why no C=C component is considered in the structures?

12. Is the simulation using one repetitive unit accurate to understand the frontier molecular orbitals? As shown in Figure 3b, the LUMO of a BQ unit shows significant asymmetric distribution over the two C=O in para-position. The lack of consideration on extended structure may oversimplified the underlying processes.

13. The author stated that the “new peaks in the TQBQ-COF electrode after recharged to 3.7 V” may suggest the cathode SEI (Figure S19). The component at 1070 eV is Na. The author should give explanations on the component peaking at ~500 eV.

14. The TQBQ-COF shows a b value closing to 1 (Figure S21), suggesting a surface control process and thus lead to the pseudo-capacitance like capacity for Na storage. Therefore, an estimation/measurement of the Na⁺ diffusion coefficient in the TQBQ-COF is necessary.

15. How about the self-discharge performance of the TQBQ-COF battery?

Reviewer #3 (Remarks to the Author):

This manuscript represents sodium storage properties of covalent organic framework from triquinoxalynylene and benzoquinone (TQBQ-COF). Discharge capacity of the resulting TQBQ-COF was very high and its rate capability was also enhanced, compared to ordinary COFs. However, many questions are remained as unclear in terms of data interpretation.

1) Authors insisted that the enhanced sodium storage of TQBQ-COF was due to its microporosity. However, according to the N₂ adsorption-desorption measurement of heat-treated TQBQ-COF (Fig. 1e), isotherm of TQBQ-COF is not a type I, based on the BDDT (Brunauer, Deming, Deming and Teller) classification. I think that most of void spaces in TQBQ-COF came from mesopores, not from micropores. Alpha-s plot or t-plot analyses should be added to the paper for elucidating microporous properties of TQBQ-COF. Pore size distribution curve in the inset of Fig 1e might be from Barrett-Joyner-Halenda (BJH) formula, which cannot explain the microporous property of TQBQ-COF. And, most of discussion on porosity should be revised, based on the accurate data interpretation.

2) Authors measured electrochemical impedance spectroscopy (EIS) of TQBQ-COF, discussing that the

enhanced rate capability and stability of the product were due to the low charge transfer resistance (R_{ct}). However, according to the Cottrell equation ($i = avb$), b value is about 1, showing that the energy storage mechanism can be explained by a capacitive process. In this case, ionic conductivity of the product can be one of the important factors determining electrochemical sodium storage properties. Therefore, discussion on ionic conductivity will be helpful for readers studying COFs.

3) In the elemental analysis (EA), thermogravimetric analysis (TGA) and N_2 adsorption measurements, small molecules attached on the COF were referred. Because the electrochemical property is affected by the structure of COF, it will be better for readers to know what small molecules were attached in the pores of COF.

4) (Supplementary Figure S4) In the TEM image of S4e, Authors determined d -spacing of (0 -1 1) plane. However, lattice fringe in Fig. S4e might be due to the lamella structure of the product, especially due to (0 0 2) planes.

Minor things

(typos) carbonthyls  carbonyl (in page 4)

Point-by-point response to the reviewers' comments

Reviewer #1:

Comments: This manuscript reports an organic framework cathode for sodium-ion batteries. This new cathode shows a high gravimetric energy density of 463.2 mAh/g. The contents are of great

interests however the manuscript does not sufficiently provide data enough to convince advantages of their newly synthesized cathode material compared to present oxide-based electrode materials. Therefore, this referee requests the authors to show following data and discussions in order to assess further whether the content of manuscript can be recommended to publish in this journal.

1. Characterizations are not enough. First of all, please bring Fig. S4 in the main manuscript. This data is very important. This referee believes that the authors should improve this data. First on Fig. S4a, there are a lot of mismatching between the simulation and the experimental pattern. The authors should show residual and discuss about the reason for difference in their model and the prepared frameworks. Furthermore, Figure S4d is quite unclear data. There is no scale, no electron beam direction and no indication which point is corresponding (002).

Reply: These have been considered. We have improved the data of PXRD and HRTEM in Fig. 1. The most contents of Fig. S4 have been brought in Fig. 1 in the revised manuscript. To clearly show the mismatching peaks between the simulation and the experimental pattern, we marked the residual peaks in red in PXRD pattern (Fig. 1c, page 7). The reason for difference in our pristine model (AB stacking) and the prepared framework is that there is a small amount of AA stacking model existing in the as-prepared TQBQ-COF. As shown in Fig. 1c, the main peaks in the experimental XRD pattern are in well agreement with the AB stacking model, and the residual peaks mostly correspond to the AA stacking model. The detailed discussions about this issue have been added in the revised manuscript. Please see lines 11-15, 20-22 (page 6), and lines 1-6 (page 7). In addition, pristine Fig. S4e is unclear and cannot indicate the point which corresponds to (002). Thus, we delete the unclear data. Meanwhile, to reveal that the point is corresponding to (002) convincingly, we have added the corresponding selected area electron diffraction (SAED) pattern in Fig. 1e which can prove the statement.

2. Please measure the conductivity of the material. The particle size is quite large (ca 1 micro meter?), therefore conductivity plays a key role for electrochemical properties.

Reply: Yes, you are right. This has been considered. The particle size is about 1 micro meter and the conductivity plays a key role for electrochemical properties. Thus, we measured both the electronic and ionic conductivity of the material. (1) The electronic conductivity of TQBQ-COF material is $1.973 \times 10^{-9} \text{ S cm}^{-1}$, which was measured by the linear sweep voltammetry measurement (Fig. S11, page S12). Compared with the most organic electrode materials, TQBQ-COF exhibits a good electronic conductivity. (2) The ionic conductivity of the discharged TQBQ-COF electrode

(including 11 wt% Super P) is $5.53 \times 10^{-4} \text{ S cm}^{-1}$, which was tested by the electrochemical impedance frequency response method in Fig. S12b (*Electrochem. Solid-State Lett.* 2007, **10**, A65-A69). Note that the ionic conductivity of as-prepared TQBQ-COF is not measured because it does not contain Na^+ . The high ionic conductivity is helpful for facile Na^+ transport, contributing to improve the rate capability of TQBQ-COF electrodes. Please see lines 12-22 (page 10) and line 1 (page 11).

3. Why graphene's D and G bands are same for this organic framework's bands? There is a robust theory to calculate the position of G band, and from this point this referee are not convinced why this material shows its G band at an identical position to graphene.

Reply: We have corrected the corresponding description. The prepared TQBQ-COF has a 2D structure which is similar to graphene, but the positions of D and G bands of the prepared TQBQ-COF show no direct correlation with that of graphene. In addition, the HRTEM and AFM results (Fig. 1d and Fig. 2b) could provide positive evidences for a "2D layer structure" for TQBQ-COF, based on which we achieve the D and G bands for the 2D TQBQ-COF layers. The G band in Raman spectrum is assigned to the sp^2 C structure, and the D band is caused by the sp^3 C and bent sp^2 C structures (*Phil. Trans. R. Soc. Lond. A* 2004, **362**, 2477-2512; *Angew. Chem. Int. Ed.* 2018, **37**, 9443-9446). The position of G band is highly relevant to the 2D structure of different materials, showing no necessarily identical position as graphene (*ACS Nano* 2019, **13**, 3600-3607; *Nat. Commun.* 2013, **4**, 1485). Please see lines 9-18 (page 9).

4. The referee sees that the electrochemical properties section should be improved. The range to show Coulombic efficiency (%) should be from 80 to 100 %. Nobody can check how CE changing on increasing cycling number. In case of organic framework, an important point is energy efficiency (i.e. Q at discharge/Q at charge). Therefore, please show the energy efficiency. Furthermore, how is the volumetric energy density? It is much more important value than gravimetric energy density.

Reply: These have been considered. We carried out more studies and tried our best to improve the electrochemical properties section with the following points. (1) The range to show Coulombic efficiency (CE, %) has been changed to 80~102% in Fig. 5d. The CE shows negligible change (~100%) on increasing cycling number, which is important for the stable cycle of TQBQ-COF electrode. (2) The energy efficiency has been shown in Fig. S31 (page 32). The results indicate that the TQBQ-COF electrode exhibits a higher energy efficiency than many reported materials (*Chem* 2019, **5**, 1-24), showing promising application in the energy devices. (3) The volumetric energy density is $\sim 1252 \text{ Wh L}^{-1}_{\text{TQBQ-COF}}$, which has been added in the revised manuscript. As the volume of the pouch-type SIB is 1.92 cm^3 ($6 \times 8 \times 0.04$), the volumetric energy density of the total pouch-cell is calculated to be $78.5 \text{ Wh L}^{-1}_{\text{cell}}$. Please see lines 4-7 (page 22) and 1-4 (page 23).

5. About pouch cell: please show the capacity normalized by the whole mass of cell. With this, please show the amount and mass of electrolyte and sodium metal.

Reply: We added the capacity normalized by the whole mass (1.49 g) of the pouch-cell in Fig. S36 (page S38). The normalized capacity is $54.4 \text{ mAh g}^{-1}_{\text{cell}}$. The corresponding description was added in the revised manuscript. Please see lines 17-22 (page 22). Moreover, the amount of the electrolyte (4.0 g Ah^{-1}) and the mass of the sodium metal (0.25 g) are listed in Table S6 (page

S38).

Reviewer #2:

Comments: Chen et al. reported COFs-based cathode materials with a high theoretical capacity for sodium-ion batteries. This work represents the first report combining N-heteroaromatic triquinoxalinylene and benzoquinone in a single material. Synthesis, structural characterization, morphology characterization, In-situ FTIR, ex-situ XPS, DFT calculation, battery performance, were carried out. However, some issues in the work should be addressed before I could make a reasonable evaluation.

1. In Figure 1a, the author seems to be providing a unit with a $12e^-$ $12 Na^+$ reduction of the inner shell. Could the author provide an accurate structure, specifically on the bond valence?

Reply: We have provided the accurate structure after a $12e^-$ $12 Na^+$ reduction, as well as the bond valence (Fig. 4c, page 16). The structure (accurate atomic coordinates) of this molecule is given in Table S2. The DFT calculation results show that the $Na_{12}TQBQ$ -COF with 6 Na^+ outside the unit plane and 6 Na^+ inside the unit plane. The Mulliken charges of two kinds of Na atoms with different chemical environment and their adjacent N and O atoms are marked out in Fig. 4c. Furthermore, the average atomic valences of Na, O, and N in $Na_{12}TQBQ$ -COF are calculated to be +1, -1, and -0.5, respectively. Please see lines 5-13 (page 17) and Table S2 (page S20-22).

2. In Fig. S17, the structure is not a repeating unit, and therefore the calculation of theoretical capacity (515 mAh/g) is wrong because of this wrong drawing. The last four structures of reduced products are not in reduced states.

Reply: Because Fig. 4a is right and could clearly show the sodiation/desodiation process for TQBQ-COF, we deleted the wrong pristine Fig. S17. The theoretical capacity of TQBQ-COF was

calculated from the equation of $C_{th} = \frac{nF}{3.6M}$, based on Fig. 1a instead of Fig. S17. As shown in Fig. 1a, M is 624.4 g mol^{-1} , n is 12 (based on one ring unit of TQBQ-COF, which is marked inside the yellow dotted line in Fig. 1a), and F is Faraday constant (96485 C mol^{-1}). Thus, we can obtain that the theoretical capacity of TQBQ-COF is 515 mAh g^{-1} . Please see Fig. 4 (page 16).

3. In the FTIR spectra, can the authors assign the vibration mode of C=C? This mode seems to be relatively strong in the TABQ.

Reply: We could assign the vibration mode of C=C to $\sim 1600 \text{ cm}^{-1}$ in the TABQ spectrum in Fig. S1. This mode is relatively strong in TABQ but become obscure in TQBQ-COF. The reason is that the symmetric stretching vibration of C=C ($\sim 1600 \text{ cm}^{-1}$) for TQBQ-COF is likely covered by the C=O stretch mode. This cover phenomenon can be proven by ex-situ FTIR in Fig. S20 (page S18). When discharged the cell to 0.8 V, the vibration of C=O disappear and we can observe the obvious vibration peak of C=C.

4. The in-situ FTIR curves is too crowded, and the range is too narrow. Could the author put some important curves in one supporting figure? Can we know the reduction sequence for C=O and C=N according to the in-situ FTIR result?

Reply: These have been addressed. We put some important curves with wider range (1700-1200 cm^{-1}) in Fig. S19. Moreover, we added the ex-situ FTIR (Fig. S20, page S18) to further show the change of the characteristic peaks. According to the in-situ FTIR result (Fig. 3b), the peaks at 1627 cm^{-1} (C=O) and 1545 cm^{-1} (C=N) become weak gradually at the same time. Thus, we can know that the reduction for C=O and C=N occurs almost at the same time. Furthermore, the XPS and DFT calculations can also support the FTIR results. (1) XPS: The ex-situ XPS results (Fig. 3c) show that the peaks at 287.0 and 287.6 eV (C=O/C=N) became weak almost at the same time, with the peaks of C-O/C-N emerged. (2) DFT calculations: As shown in Fig. 4b, the similar percentage (~20%) of nitrogen and oxygen in the orbital composition of LUMO indicates that both nitrogen and oxygen atoms are stabilized the Na^+ ions collaboratively, which are in good accordance with the experimental FTIR and XPS tests. Please see lines 10-17 (page 12), lines 4-9 (page 14).

5. In Fig. S3, the D band in Raman spectrum may not only originated from the “defection of the graphene-like structure”. It could be caused by the sp^3 C and bent sp^2 C structures. And such results do not provide implications for a “2D layer structure”. AFM and TEM (Fig. S4 and S5) could provide more insights.

Reply: Yes, you are right. The D band in Raman spectrum is caused by the sp^3 C and bent sp^2 C structures, the G band is assigned to the sp^2 C structure, and the D and G bands are not enough to determine a “2D layer structure” (*Phil. Trans. R. Soc. Lond. A* 2004, **362**, 2477-2512; *Angew. Chem. Int. Ed.* 2018, **37**, 9443-9446). To provide implications for a “2D layer structure” for TQBQ-COF, the HRTEM and AFM results are provided as follows: (1) The TEM image (Fig. 1d) shows an interlayer distance of ~0.3 nm between the conjugated TQBQ-COF layers. (2) AFM image (Fig. 2b) suggests that the TQBQ-COF layers exist obviously with a height of ~5 nm for 16 monolayers. We revised the description regarding to the 2D layer structure of TQBQ-COF in the revised manuscript. Please see lines 9-18 (page 9).

6. Is the XPS data/peak position corrected with a standard component? Although TQBQ-COF is a n-type semiconductor, the peak positions may still suffer from deviation from correct positions.

Reply: We appreciate your significant suggestion. We are sorry for our negligence of the small difference (0.4 eV) between our XPS peak position and a standard C1s peak (*Angew. Chem. Int. Ed.* 2011, **50**, 8753-8757). We have corrected all of the XPS data/peak position with a standard C1s peak (284.5 eV) in the revised manuscript. Please see lines 4-9 (page 8), and Fig. S6 (page S8).

7. The XPS fitting results in Fig. S7 show a wide distribution of FWHM. In Fig. S7, the C 1s BE of C=N is lower than that of C=C. The author stated that the lower BE may be caused by the delocalization of lone pair electrons of N over the conjugated structure. Could the author give more solid evidence, since such low BE in C 1s is rarely observed, even in g-C3N4?

Reply: Yes. We improved the data of the C1s/N1s XPS spectra in Fig. S6 with a reasonable distribution of FWHM. Firstly, we corrected the C1s peak with the standard C1s peak (284.5 eV) according to the reviewer's above suggestion. Secondly, we reanalyzed the XPS data and confirmed that the C1s BE of C=N is 287.0 eV, which is higher than that of C=C (284.5 eV). The current C1s BE of C=N is normal and not low. We corrected the corresponding descriptions in the

revised manuscript. Please see lines 4-9 (page 8) and Fig. S6 (page S8).

8. The C=O BE in Fig. S7 is high as ~288 eV, which is not usually seen in conjugated C=O moieties. If as the author stated that N is delocalizing electron over the system, how could the BE of C 1s in C=O is even higher than the conventional observed values (~287 eV). What is the possible effect of N on the XPS of C1s spectrum? In Figure S7b, the label C=N appeared twice, and that for C-N is missing.

Reply: We have considered with the following three points. (1) After correcting the C1s peak with the standard C1s peak (284.5 eV), the binding energy of C=O peak is actually at ~287.6 eV. This value is normal due to the π -conjugated structure of TQBQ-COF and in agreement with many C=O-contained works (*Nat. Commun.* 2016, **7**, 13318; *Angew. Chem. Int. Ed.* 2011, **50**, 8753-8757; *Nano Energy* 2017, **41**, 117-127). We have revised the XPS of C1s spectrum in Fig. S6a and the ex-situ XPS results in Fig. 3c (page 13). (2) The effect of N atoms on the XPS of C1s spectrum is that the N atom has a higher electronegativity than C atom, which makes the electron cloud density of the C from C=N lower than that from C=C. Moreover, the lone pair electrons on the N atom among the conjugated structure may slightly lower the binding energy of the C1s spectrum. (3) We have corrected the N1s spectrum, in which it shows only one peak for N=C, and the missing of N-C peak can be attributed to the very small amount of the functional groups in the edge of TQBQ-COF. Please see lines 11-14 (page 13) and lines 4-9 (page 14).

9. The authors found that the as-prepared TQBQ-COF may have absorbed small molecules as evidenced by the TGA result. In my opinion, the FTIR data also suggests this hypothesis. The C-H mode at ~2900 cm^{-1} diminished significantly after heat treatment.

Reply: Yes. The TGA result and FTIR data both suggested the absorbed small molecules in the holes of the as-prepared TQBQ-COF. After considering the preparation process, we think that there are some H₂O, O₂, CH₃OH and CO₂ included in the TQBQ-COF, which can be obtained from the FTIR data in Fig. S1, where the O-H mode (3200 cm^{-1}), the CO₂ peak (2300 cm^{-1}), and the C-H mode (~2900 cm^{-1}) diminished significantly after heat treatment.

10. What is the possible reason for the high charge transfer resistance in the PC-based electrolyte? Is it due to the solvation of Na⁺ or the interaction of solvent molecules with the TQBQ-COF backbone?

Reply: The possible reason for the high charge transfer resistance in the PC-based electrolyte are as follows: (1) Taking the Na anode side into consideration (Fig. S16), the Na//Na symmetrical cell with the PC-based electrolyte shows much higher charge transfer resistance (380 Ω) than that with DEGDME-based electrolyte (8.4 Ω), indicating that the Na anode side is highly responsible for the high charge transfer resistance of TQBQ-COF//Na battery in the PC-based electrolyte. (2) The high charge transfer resistance in the PC-based electrolyte also results from the solvation effect of Na⁺. As shown in Fig. S14 (page S14), the diameter of [NaPC3]⁺ solvation configuration (7.05 Å) in PC-based electrolyte is much larger than that of [NaG22]⁺ (4.33 Å) in DEGDME-based electrolyte. The large radius of [NaPC3]⁺ would inevitably cause difficulties for Na⁺ diffusion in the PC-based electrolyte. In addition, the D_{Na^+} in TQBQ-COF electrode in the DEGDME-based electrolyte is almost same with the PC-based electrolyte (~10⁻¹¹ cm² s⁻¹) according to the CV measurements (Fig. S15 and Fig. S29), implying that the interaction of

solvent molecules with the TQBQ-COF backbone has little responsibility for the high charge transfer resistance. We added the reasons in the revised manuscript. Please see lines 12-15 (page 11).

11. In Figure 2, what is the mechanism of “the N-fused conjugated skeleton” to induce existence of a peak component at 1545 cm^{-1} at a fully discharged state? Why no C=C component is considered in the structures?

Reply: We considered this with the following points. (1) When discharged to 0.8 V, the N-fused conjugated skeleton contains C=N, which is located at 1545 cm^{-1} according to the pristine state. (2) In fact, the C=C mode is included in the N-fused conjugated skeleton. Affected by the influence from the electrolyte or the testing condition, the peak intensity of C=C seems to be covered in the in-situ FTIR. This cover phenomenon can be proven by ex-situ FTIR in Fig. S20 (page S18). When discharged the cell to 0.8 V, the vibration of C=C can be observed at around 1600 cm^{-1} . Please see lines 10-17 (page 12).

12. Is the simulation using one repetitive unit accurate to understand the frontier molecular orbitals? As shown in Figure 3b, the LUMO of a BQ unit shows significant asymmetric distribution over the two C=O in para-position. The lack of consideration on extended structure may oversimplified the underlying processes.

Reply: Yes. The simulation is using one repetitive unit accurate to understand the frontier molecular orbitals. The asymmetric distribution of LUMO results from the different chemical environment of the C=O groups in para-position in one single TQBQ-COF unit ring. However, we still believe that these C=O groups indeed have the same priority in accommodating Na^+ ions, for their chemical environment are exactly the same in a periodic system. To confirm this, we have calculated the MESP and LUMO patterns of two TQBQ-COF repetitive units, as shown in Fig. S21 (page S19). MESP on the two-unit TQBQ-COF surface minima indicates the same affinity of the para-positioned C=O groups to Na^+ ions, and the symmetrical attribution of LUMO pattern on the C=O groups in the red circle also exhibit the same reaction activity. Therefore, we consider that studying only the six C=O groups at the inner side of one repetitive ring unit is sufficient for simulating all C=O groups in an extended structure of TQBQ-COF. Please see lines 10-14 (page 15).

13. The author stated that the “new peaks in the TQBQ-COF electrode after recharged to 3.7 V” may suggest the cathode SEI (Figure S19). The component at 1070 eV is Na. The author should give explanations on the component peaking at $\sim 500\text{ eV}$.

Reply: The peak at $\sim 500\text{ eV}$ belongs to the corresponding auger peak of Na1s (*ACS Nano* 2019, DOI:10.1021/acsnano.9b03492). To make the XPS data clearer for readers, we have marked all of the elements and their corresponding auger peaks in Fig. S26 (page S28) to further explain the cathode SEI components.

14. The TQBQ-COF shows a b value closing to 1 (Fig. S21), suggesting a surface control process and thus lead to the pseudo-capacitance like capacity for Na storage. Therefore, an estimation/measurement of the Na^+ diffusion coefficient in the TQBQ-COF is necessary.

Reply: Yes, you are right. It is of importance to measure Na^+ diffusion coefficient in the

TQBQ-COF to explain the pseudo-capacitance like capacity for Na storage. We used both CV and GITT (*Chem* 2018, **4**, 2600-2614; *J. Electroanal. Chem.* 1997, **421**, 79-88) to measure the Na⁺ diffusion coefficient (D_{Na^+}). The two measurements all reveal that the D_{Na^+} is about 10^{-11} cm² s⁻¹. The high D_{Na^+} is beneficial for fast charge and discharge of TQBQ-COF electrode. We have added the CV and GITT measurements of the Na⁺ diffusion coefficient in the revised manuscript. Please see Fig. S29 and S30 (page S31), lines 14-22 (page 20) and lines 1-15 (page 21).

15. How about the self-discharge performance of the TQBQ-COF battery?

Reply: We have tested the self-discharge performance of the TQBQ-COF battery after standing the fully charged battery for 24 hours. The results reveal that 97.2% of the original capacity of the TQBQ-COF battery retained (Fig. S27, page S29), which can be attributed to the good stability of the TQBQ-COF material and few side reactions. Please see lines 3-8 (page 19).

Reviewer #3:

Comments: This manuscript represents sodium storage properties of covalent organic framework from triquinoxalinyne and benzoquinone (TQBQ-COF). Discharge capacity of the resulting TQBQ-COF was very high and its rate capability was also enhanced, compared to ordinary COFs. However, many questions are remained as unclear in terms of data interpretation.

1) Authors insisted that the enhanced sodium storage of TQBQ-COF was due to its microporosity. However, according to the N₂ adsorption-desorption measurement of heat-treated TQBQ-COF (Fig. 1e), isotherm of TQBQ-COF is not a type I, based on the BDDT (Brunauer, Deming, Deming and Teller) classification. I think that most of void spaces in TQBQ-COF came from mesopores, not from micropores. Alpha-s plot or t-plot analyses should be added to the paper for elucidating microporous properties of TQBQ-COF. Pore size distribution curve in the inset of Fig 1e might be from Barrett-Joyner-Halenda (BJH) formula, which cannot explain the microporous property of TQBQ-COF. And, most of discussion on porosity should be revised, based on the accurate data interpretation.

Reply: The N₂ adsorption-desorption isotherm of TQBQ-COF is not a typical type I (Fig. 2c), and it reveals that most of void space in TQBQ-COF comes from mesopores, not from micropores. Pore size distribution curve in the inset of Fig. 1e is actually from Barrett-Joyner-Halenda (BJH) formula, and we get the HK-Plot based on it. Moreover, we added the t-Plot analyses in the inset of Fig. 2c, which shows a model for microporous structure of TQBQ-COF. Combining the t-Plot and the N₂ adsorption-desorption measurement results, we can know that most pores inside TQBQ-COF are mesoporous, with some micropores inside. We have revised the discussion on porosity in the revised manuscript, and used 'mesopore' to depict the pore property of TQBQ-COF. Please see lines 2-7 (page 10).

2) Authors measured electrochemical impedance spectroscopy (EIS) of TQBQ-COF, discussing that the enhanced rate capability and stability of the product were due to the low charge transfer resistance (Rct). However, according to the Cottrell equation ($i = a v^b$), b value is about 1, showing that the energy storage mechanism can be explained by a capacitive process. In this case, ionic conductivity of the product could be one reason to enhance rate capability and stability of the product. Therefore, discussion on ionic conductivity will be helpful for readers studying COFs.

Reply: Thank you for your suggestion. The ionic conductivity of the discharged TQBQ-COF (including 11 wt% Super P) is $5.53 \times 10^{-4} \text{ S cm}^{-1}$. The detailed discussion can be seen in the Reply to question 2 of Reviewer 1.

3) In the elemental analysis (EA), thermogravimetric analysis (TGA) and N_2 adsorption measurements, small molecules attached on the COF were referred. Because the electrochemical property is affected by the structure of COF, it will be better for readers to know what small molecules were attached in the pores of COF.

Reply: The small molecules attached in the pores of COF are H_2O , O_2 , CH_3OH and CO_2 , which has been proven by FTIR data (Fig. S1). The analysis process can be seen in the Reply to question 9 of Reviewer 2. Because these small molecules attached on the COF may affect the electrochemical performance, we heated the samples at $200 \text{ }^\circ\text{C}$ under vacuum to remove these small molecules before use.

4) (Supplementary Figure S4) In the TEM image of S4e, Authors determined d-spacing of (0-11) plane. However, lattice fringe in Fig. S4e might be due to the lamella structure of the product, especially due to (002) planes.

Reply: Yes. You are right. The distinct interlayer distance of $\sim 0.3 \text{ nm}$ in Fig. 1d is assigned to the (002) facet of TQBQ-COF, corresponding to the simulated AB stacking model. The lattice fringe in original Fig. S4e might be due to the lamella structure of (002) plane of TQBQ-COF layers. Therefore, it is not reasonable to determine the d-spacing of (0-11) plane from the original Fig. S4e, and we deleted the unclear illustration in original Fig. S4. The peak at 15.69° in PXRD pattern could be assigned to the d-spacing of (0-11) plane, which is in agreement with the pore size (5.6 \AA) of the AB stacking model of TQBQ-COF layers. We revised our unclear sentences and improved our experiment in the revised manuscript. Please see lines 11-15 (page 6).

Minor things

(typos) carbonthyls  carbonyl (in page 4)

Reply: The typos have been corrected in page 5 (line 6). Moreover, we tried our best to correct all the grammatical mistakes, ranging from typos, syntactical inconsistencies, misplaced tenses etc. throughout our manuscript. All the changes are marked with blue fonts.

Meanwhile, there are some other changes in the revised manuscript. All the changes have been marked with blue fonts. We hope that the revised manuscript is meeting the requirements. Thank you again.

REVIEWERS' COMMENTS:

Reviewer #1 (Remarks to the Author):

The authors made significant improvements and this referee was convinced by the contents of this manuscript. Therefore this referee recommends the editor to accept this manuscript for publication as it is.

Reviewer #2 (Remarks to the Author):

The revision has addressed my questions.

Reviewer #3 (Remarks to the Author):

Most of issues raised by reviewers were addressed properly, and this article could be recommended to publish in Nature Communications

Reply to Reviewers' comments:

Reviewer #1 (Remarks to the Author):

The authors made significant improvements and this referee was convinced by the contents of this manuscript. Therefore, this referee recommends the editor to accept this manuscript for publication as it is.

Reply: We do thank the reviewer's positive evaluation.

Reviewer #2 (Remarks to the Author):

The revision has addressed my questions.

Reply: We greatly appreciate the reviewer's important comments for improving our paper.

Reviewer #3 (Remarks to the Author):

Most of issues raised by reviewers were addressed properly, and this article could be recommended to publish in Nature Communications.

Reply: We highly appreciate your positive evaluation. Thank you for the significant suggestions for improving our work.